

# RNA sequencing (RNA-Seq) of lymph node, spleen, and thymus transcriptome from wild Peninsular Malaysian cynomolgus macaque (*Macaca fascicularis*)

Joey Ee Uli[1], Christina Seok Yien Yong[2], Swee Keong Yeap[3], Jeffrine J. Rovie-Ryan[4], Nurulfiza Mat Isa[1], Soon Guan Tan[1] and Noorjahan Banu Alitheen[1]

[1] Department of Cell and Molecular Biology, Faculty of Biotechnology and Biomolecular Sciences, Universiti Putra Malaysia, Serdang, Selangor, Malaysia
[2] Department of Biology, Faculty of Science, Universiti Putra Malaysia, Serdang, Selangor, Malaysia
[3] China-ASEAN College of Marine Sciences, Xiamen University, Sepang, Selangor, Malaysia
[4] Department of Wildlife and National Parks (DWNP), Ex-Situ Conservation Division, Department of Wildlife and National Parks, Kuala Lumpur, Malaysia

Corresponding author
Noorjahan Banu Alitheen, noorjahan@upm.edu.my

## ABSTRACT

The cynomolgus macaque (*Macaca fascicularis*) is an extensively utilised nonhuman primate model for biomedical research due to its biological, behavioural, and genetic similarities to humans. Genomic information of cynomolgus macaque is vital for research in various fields; however, there is presently a shortage of genomic information on the Malaysian cynomolgus macaque. This study aimed to sequence, assemble, annotate, and profile the Peninsular Malaysian cynomolgus macaque transcriptome derived from three tissues (lymph node, spleen, and thymus) using RNA sequencing (RNA-Seq) technology. A total of 174,208,078 paired end 70 base pair sequencing reads were obtained from the Illumina Hi-Seq 2500 sequencer. The overall mapping percentage of the sequencing reads to the *M. fascicularis* reference genome ranged from 53–63%. Categorisation of expressed genes to Gene Ontology (GO) and KEGG pathway categories revealed that GO terms with the highest number of associated expressed genes include Cellular process, Catalytic activity, and Cell part, while for pathway categorisation, the majority of expressed genes in lymph node, spleen, and thymus fall under the Global overview and maps pathway category, while 266, 221, and 138 genes from lymph node, spleen, and thymus were respectively enriched in the Immune system category. Enriched Immune system pathways include Platelet activation pathway, Antigen processing and presentation, B cell receptor signalling pathway, and Intestinal immune network for IgA production. Differential gene expression analysis among the three tissues revealed 574 differentially expressed genes (DEG) between lymph and spleen, 5402 DEGs between lymph and thymus, and 7008 DEGs between spleen and thymus. Venn diagram analysis of expressed genes revealed a total of 2,630, 253, and 279 tissue-specific genes respectively for lymph node, spleen, and thymus tissues. This is the first time the lymph node, spleen, and thymus transcriptome of the Peninsular Malaysian cynomolgus macaque have been sequenced via RNA-Seq. Novel transcriptomic data will further enrich the present *M. fascicularis* genomic database and provide future research potentials, including novel transcript discovery, comparative studies, and molecular markers development.

# INTRODUCTION

The cynomolgus macaque (*Macaca fascicularis*), also known as the long-tailed macaque or crab-eating macaque, are nonhuman primates (NHP) belonging to the Cercopithecidae family. They occur in the southeastern region of Bangladesh, the mainland of Southeast Asia (including Malaysia, Laos, Cambodia, Vietnam, and Thailand), the islands of Indonesia, Borneo, and the Philippines and the Nicobar Islands of India (*Groves, 2001*), and more recently Mauritius (*Ferguson et al., 2007*). Out of the 50 *Macaca fascicularis* subspecies presently classified, the *Macaca fascicularis fascicularis* subspecies is the most widely distributed macaque subspecies in Peninsular Malaysia (*Abdul-Latiff et al., 2014*), mostly occurring in mangrove forests, lowland rainforests, and in the peripheral of urban dwellings.

Among the myriad of model organisms used in biomedical research, the cynomolgus macaque have become one of the most heavily utilised NHP models due to their close evolutionary relationship with humans (*Perelman et al., 2011*). Biological, physiological, behavioural, and genetic similarities between humans and cynomolgus macaques make these macaques capable of recapitulating symptoms of diseases observed in humans (*Patel, Jhamb & Singh, 2011*; *Shively et al., 2015*). As a result, the cynomolgus macaques are regarded as suitable model organisms for translational studies in the biomedical field (*Carlsson et al., 2004*). Their inclusion in numerous immunological, neuroscience, vaccine development, and pharmacokinetic studies have shown the cynomolgus macaque's versatility as NHP model organisms (*Higgs & Ziegler, 2010*; *Nunamaker et al., 2013*; *Lee et al., 2014b*; *Berry et al., 2015*). In order to benefit the biomedical field, complete genomic and transcriptomic information has become essential in expediting the understanding of gene expressions and biological pathways relevant to studies of interest. Aside from biomedicine, complete genomic and transcriptomic information enables researchers to perform phylogenomic studies of relevant organisms. Aligning entire genomes enables a higher base resolution to infer phylogeny. High throughput sequencing serves as a tool to obtain genomic and transcriptomic information of any organism of interest from relatively small starting material and at a fraction of the cost involved in Sanger and capillary shotgun sequencing approaches. In recent years, RNA sequencing (RNA-Seq) has become an indispensable method to sequence entire transcriptomes. Downstream applications of RNA-Seq include differential gene expression analyses, expression profiling, novel transcript discovery, single nucleotide polymorphism (SNP) discovery, and also development of molecular markers for population genetics studies (*Han et al., 2015*).

In the past decade, efforts have been made to sequence the genome and transcriptome of the cynomolgus macaque from various locations using the high-throughput sequencing approach (*Yan et al., 2011*; *Higashino et al., 2012*; *Osada et al., 2015*). To date, the transcriptome of the Malaysian cynomolgus macaque has yet to be sequenced, and while the transcriptomes of the lymph node, spleen, and thymus tissues harvested from cynomolgus

macaques have already been sequenced via various high-throughput sequencing platforms, the individuals sequenced were of Mauritian, Vietnamese, Chinese, and Philippine origin (*Ebeling et al., 2011*; *Huh et al., 2012*; *Lee et al., 2014a*; *Peng et al., 2014*). In addition, to the best of our knowledge, the macaque transcriptomes sequenced in these previous studies were individuals raised in laboratory conditions. While it is beneficial for biomedical research that model organisms like cynomolgus macaques are bred in controlled laboratory conditions to minimise genetic and phenotypic variations, it ultimately does not reflect the variable environment in which humans live (*Pedersen & Babayan, 2011*). Comparisons between wild and laboratory strains of *Mus musculus* have shown the wild strain to possess more variable immune responses compared with the laboratory strain (*Abolins et al., 2011*).

Furthermore, genetic variations among different populations of NHP models have resulted in varying phenotypical reactions to treatments in biomedical research. One such observation was made in the Indian and Chinese rhesus macaques (*Macaca mulatta*), whereby Chinese rhesus macaques exhibited slower progression of acquired immune deficiency syndrome (AIDS)-like viral infection compared with Indian rhesus macaques (*Trichel, Rajakumar & Murphey-Corb, 2002*). Prior to conducting studies utilising NHP models, it is therefore necessary to study their genetic background as NHP models originating from various geographical locations exhibit inter- and intraspecific genetic variations (*Haus et al., 2014*). Phylogenetic studies of the various populations of cynomolgus macaques across Indochina and Southeast Asia, have suggested the division of the cynomolgus macaques into two groups—namely the continental group consisting of Indochinese and Malaysian populations, and the insular group comprising of Indonesian, Bornean, and Philippine populations (*Tosi & Coke, 2007*; *Blancher et al., 2008*; *Rovie-Ryan et al., 2013*). Based on mitogenomic data, further subdivisions were observed within the continental group that separates the Indochinese and Malaysian macaque populations (*Liedigk et al., 2015*). Owing to how the Peninsular Malaysian cynomolgus macaque have higher levels of nucleotide diversity and are monophyletic compared with other Southeast Asian cynomolgus macaque populations (*Smith, McDonough & George, 2007*; *Abdul-Latiff et al., 2014*), we believe additional transcriptomic information of the wild Malaysian cynomolgus macaque will augment the existing cynomolgus macaque transcriptomic dataset for future biomedical researches, for instance, studies related to AIDS which rely on panels of immune function-related genes that are potentially expressed in lymph node, spleen, and/or thymus tissues.

In this present study, we sequence three tissues–lymph node, spleen, and thymus, harvested from cynomolgus macaques exposed to urban environments using the Illumina HiSeq 2500 platform. A reference-guided assembly was performed to assemble the sequencing reads by mapping the reads to the *M. fascicularis* reference genome, followed by empirical analysis of differential gene expression (EDGE test), and functional annotations and categorisation of the assembled reads. The dataset presented here will prove valuable for future immunological or drug translational studies involving cynomolgus macaques as NHPs.

## MATERIALS AND METHODS

### Ethics statement

The sampling and euthanasia of conflict *M. fascicularis* macaques were carried out by the Department of Wildlife and National Parks (DWNP) Malaysia (Permit No: JPHL&TN(IP): 80-4/2) and were performed according to the guidelines set by the Institutional Animal Care and Use Committee (IACUC), University of California, Davis, United States of America as adopted by the PREDICT Project in Malaysia. Permission to obtain macaque lymph node, spleen, and thymus tissue samples for research purposes by the Genetics Lab at the Department of Biology was granted by DWNP.

### *Macaca fascicularis* sampling

Three male conflict *M. fascicularis* individuals from the state of Selangor in Malaysia were captured by DWNP. The three individuals were adolescents with total length of 79.0 cm, 80.5 cm, and 91.0 cm and weighing at 1.63 kg, 1.43 kg, and 2.11 kg respectively, and appeared to belong in the same family group. Based on visual examination, the macaques were free of disease. Lymph node, spleen, and thymus tissues of the three individual macaques were harvested and stored separately in 1.5 ml tubes filled with RNAlater RNA Stabilization Agent (Qiagen, Hilden, Germany), which were incubated in 4 °C overnight, and subsequently stored in −80 °C.

### RNA extraction & quality check

Total RNA samples were extracted from the nine harvested tissues using Qiagen's RNeasy Mini Kit. Due to the presence of genomic DNA in the RNA extract, a modified protocol was employed by utilising Epicentre's DNase I solution to remove genomic DNA contamination. The intactness of the RNA extracts were visualised using 1% agarose gel, and their respective concentrations quantitated using a Qubit fluorometer (Life Technologies, Carlsbad, CA, USA). The integrity of the RNA extracts were determined using Agilent's 2100 Bioanalyzer (Agilent Technologies, Inc., Santa Clara, CA, USA) via an RNA Pico chip. RNA samples with RIN of more than 7.0 were selected for the subsequent library preparation step.

### RNA-Seq library preparation & sequencing

A total of nine sequencing libraries were prepared, with three replicates for each tissue type. The library preparation step consisted of two main sections; the removal of ribosomal RNA (rRNA) from the total RNA, and the conversion of the rRNA-depleted RNA to cDNA. rRNA depletion was carried out using the Ribo-Zero™ Gold Kit (Human/Mouse/Rat) (Epicentre, Madison, WI, USA), while the conversion of treated RNA to cDNA was performed using the ScriptSeq™ v2 RNA-Seq Library Preparation Kit (Epicentre, Madison, WI, USA). The size fragmentation distribution of the constructed cDNA library was assessed using Agilent's 2100 Bioanalyzer (Agilent Technologies, Santa Clara, CA, USA) via a High-Sensitivity DNA assay. The cDNA library was also subjected to quantitation via a RT-qPCR assay using a CFX96 Touch™ Real-Time PCR Detection System (Bio-Rad, Hercules, CA, USA). The final normalised cDNA library was pooled together in a single reaction and was sequenced

on Illumina's HiSeq 2500 platform in rapid run mode for $75 \times 2$ bp paired-end reads at the High Impact Research Centre in Universiti Malaya, Malaysia. Paired end sequencing data for lymph node, spleen, and thymus were submitted to NCBI Short Read Archive (SRA) database under accession SRP096937.

## Data analysis

Base quality and sequencing adapter trimming, reads mapping, biological replicates correlation analyses, and subsequent differential gene expression analyses were carried out on CLC Genomics Workbench v.7.5.1 (Qiagen, København, Denmark). For base quality checking and sequence adapter trimming, quality limit $= 0.01$, or the equivalent of $Q_{20}$, was utilised to remove bases with a quality value of less than 20 as well as sequencing adapters. The filtered reads were then mapped to the *M. fascicularis* reference genome (accession number: GCF_000364345.1) via reference guided assembly. Reads were set to map to "Gene Regions Only", including intron and exon regions of the genome. Expression levels were calculated as total number of reads mapped to a gene (Total Gene Reads).

A suite of biological replicates correlation analyses was performed, including normalisation, hierarchical clustering of samples, principal component analysis, and box plot analysis. Normalisation by total reads per million was performed using CLC Genomics Workbench's normalisation suite, and is similar to the counts per million (CPM)/transcripts per million (TPM) method of normalising RNA-Seq count data. Subsequent analyses including hierarchical clustering of samples, principal component analysis, and box plot analysis were performed using normalised expression values to determine the consistency of the gene expression profiles across replicates and tissue samples. For hierarchical clustering of samples, 1-Pearson correlation distance measure was employed together with average cluster linkage. Scatter plots of the gene expression profiles between biological replicates were generated for the principal component analysis based on normalised expression values (Figs. S1–S6). Pearson correlation values ($r$-value) were generated from the scatter plot.

## Differential gene expression analysis

Empirical analysis of differentially expressed genes (EDGE test) was carried out on normalised expression profiles to identify differentially expressed genes between the lymph node, spleen, and thymus tissue samples. Three separate tissue comparisons were performed—spleen vs. lymph, thymus vs. lymph, and spleen vs. thymus. Differentially expressed genes were filtered by selecting for genes with normalised fold change values of $>2$ or $<-2$. Selection of significantly differentially expressed genes were performed by filtering for genes with FDR corrected $p$-value $<0.05$.

## Gene ontology and KEGG pathway annotation & categorisation

Genes were assigned Gene Ontology and KEGG pathway annotations by applying gene ontology and pathway association files (.gaf2) obtained from the UniProtKB and KEGG databases respectively. Functional and pathway categorisation of expressed genes (normalised expression value $>1$) were carried out using a suite of online databases. GO categorisation was carried out via Panther Database version 11.1 released on
2016-10-24 (http://pantherdb.org/) utilising *Macaca mulatta* as the reference organism. Classifications of genes into biological process, molecular function, and cellular component GO categories were visualised via the "Functional classification viewed in pie chart" analysis. For KEGG pathway categorisations, DAVID Bioinformatics Resources version 6.8 (https://david-d.ncifcrf.gov/home.jsp) was utilised with the *M. fascicularis* as the background organism. EASE threshold score was set at 0.01 when determining significantly enriched pathway categories. KEGG pathway mappings with highlighted mapped objects were retrieved from KEGG mapper version 2.8 tool in the KEGG website (http://www.kegg.jp/kegg/mapper.html).

### Tissue-specific genes

To identify and visualise tissue-specific genes in lymph node, spleen, and thymus tissues, expressed genes for each tissue type were input to the web tool Venn Diagrams (http://bioinformatics.psb.ugent.be/webtools/Venn).

### RNA-seq data validation

Validation of RNA-Seq gene expression profiles were performed using NanoString nCounter XT gene expression assay (NanoString Technologies Inc., Seattle, WA, USA). A total of 24 genes were selected for validation based on fold change patterns and immune-related functions. An additional three housekeeping genes (*ACTB*, *GAPDH*, and *YWHAZ*) were also selected for reference genes normalising of digital counts data. Fold change patterns for 24 genes in their respective tissue comparisons were compared between the RNA-Seq and NanoString platforms. Custom CodeSet probes were designed from *M. fascicularis* FASTA sequences obtained from the reads mapping assembly from this study. Table S1 lists the genes utilised for the RNA-Seq data validation and their respective probe pair sequences. The amount of total RNA utilised for the hybridisation step was 200 ng. Digital scanning of RNA-probe hybrids were performed at High 280 Field of View (FOV) setting. Hybridisation and digital scanning of the RNA-probe hybrids were performed at High Impact Research Centre in Universiti Malaya, Malaysia.

## RESULTS

### Sequence reads filtering & mapping

We investigated the transcriptome of three tissues in triplicates obtained from the Malaysian *M. fascicularis* using the RNA-Seq platform. The number of raw sequence reads obtained from the high-throughput sequencing of lymph node, spleen, and thymus tissues were 50,180,478, 63,956,392, and 60,071,208 respectively with a sequence length of 70 base pairs for each read. Raw sequence reads were trimmed and filtered to remove PhiX Illumina positive control sequences, adapter and index barcode sequences, and low quality sequences based on base quality $Q_{20}$. Post-trimming, the number of sequence reads for lymph node, spleen, and thymus tissues were 47,559,293, 60,285,505, and 56,583,352 respectively, with the average length of the sequence reads ranging from 57.5 base pairs to 60.3 base pairs. The percentage of reads that were retained post-trimming ranged from 96.52% to 97.73%, indicating that at least 96% of the reads were above base quality $Q_{20}$ and were suitable for reference-guided assembly.

**Table 1** Filtering and mapping statistics of lymph, spleen, and thymus sequence reads to *Macaca fascicularis* reference genome (GCF_000364345.1). Reads were mapped to gene regions only with a maximum number of hits for a read = 10. Only intact paired reads were taken into account when counting the fragments by type.

| | Tissue | | |
| | Lymph | Spleen | Thymus |
|---|---|---|---|
| Reads (Before trim) | 50,180,478 | 63,956,392 | 60,071,208 |
| Reads (After trim) | 47,559,293 | 60,285,505 | 56,583,352 |
| Reads uniquely mapped to exon regions | 9,712,306 | 14,061,540 | 9,520,236 |
| Reads uniquely mapped to intron regions | 7,508,538 | 10,242,700 | 6,674,232 |
| Reads mapped in pairs (%) | 38.44 | 42.66 | 31.47 |
| Reads mapped in broken pairs (%) | 15.13 | 19.60 | 22.93 |
| Overall mapping percentage (%) | 53.57 | 62.26 | 54.40 |

The filtered reads from each replicate were then mapped to 36,233 genes within the *M. fascicularis* reference genome obtained from NCBI (Accession: GCF_000364345.1). The total number of paired and broken paired reads mapped to the reference genome for lymph, spleen, and thymus samples are 25,476,162, 37,536,104, and 30,782,715 reads respectively. For the counting of fragments mapped to exon and intron regions, only paired end reads were taken into account. The overall mapping percentage (sum of percentage of reads mapped in pairs and broken pairs) ranged from 53.00% to 63.00% with lymph showing lower overall mapping percentage compared to spleen and thymus. Table 1 summarises the filtering and mapping statistics of lymph node, spleen, and thymus sequence reads. Supplementary information on the individual mapping statistics for each individual library are included in Table S2.

## Biological replicates correlation analysis

Biological replicates correlation analysis of lymph node, spleen, and thymus tissues show consistent expression values in the three replicates for each tissue type (Pearson correlation value, $r > 0.8$). No outliers were detected and excluded from the subsequent analyses. Principal component analysis scatter plots and box plots of the normalised expression values were also generated and included in (Figs. S1–S6).

## Differential gene expression analysis

Three tissue comparison experiments were set up, namely Spleen vs. Lymph, Thymus vs. Lymph, and Spleen vs. Thymus. The total of differentially expressed genes called for Spleen vs. Lymph, Thymus vs. Lymph, and Spleen vs. Thymus were 574, 5,402, and 7,008 genes respectively. In Spleen vs. Lymph comparison, *CCL17* and *AIF1L* were among the top regulated genes in lymph and spleen respectively. For the Thymus vs. Lymph tissue comparison, the genes *CCR7* and *PRH2* were the top regulated genes in lymph and thymus respectively. Lastly, for Spleen vs. Thymus comparison, *MYO7A* and *KLK1* were among the top highest regulated genes for spleen and thymus respectively. Table S3 lists the differentially expressed genes with their annotated gene names and descriptions.

**Table 2** Total number of genes with Gene Ontology (Biological Process, Molecular Function, and Cellular Component) and KEGG pathway annotations.

| Annotation category | Gene ontology | | | KEGG pathway |
|---|---|---|---|---|
| | Biological process | Molecular function | Cellular component | |
| Number of annotated genes | 12,626 | 12,637 | 13,230 | 5,387 |

## Functional annotations and categorisation

To assign Gene Ontology (GO) and KEGG pathway annotations to the genes, gene ontology and pathway annotation files (.gaf2) were obtained from UniProtKB and KEGG database respectively and imported to CLC Genomics Workbench. The numbers of genes with gene ontology and pathway annotations are listed in Table 2.

To classify the genes into GO categories, expressed genes were input to the Panther Database. For lymph node tissue, 15,611, 7,876, and 6,141 genes were categorised to more than one GO categories in the Biological Process, Molecular Function, and Cellular Component domains respectively. In the spleen tissue, 14,838, 7,465, and 5,892 genes were categorised to more than one GO categories in the Biological Process, Molecular Function, and Cellular Component domains respectively. While in the thymus tissue, 12,161, 6,143, and 5,029 genes were categorised to more than one GO categories in the Biological Process, Molecular Function, and Cellular Component domains respectively. The distribution of the genes into different level 2 GO categories are shown in Table 3. The Biological Process category with the highest number of expressed genes for all three tissues is Cellular Process (Lymph, 4,429; Spleen, 4,239; Thymus, 3,551). For Molecular Function GO domain, the Catalytic Activity category contains the highest number of expressed genes for all three tissues (Lymph, 2,972; Spleen, 2,856; Thymus, 2,411). As for Cellular Component, all three lymph, spleen, and thymus tissues show the highest number of expressed genes in the Cell Part category (Lymph, 2,339; Spleen, 2,247; Thymus, 2,002).

Expressed genes were also input to DAVID Bioinformatics Resources to classify the genes into KEGG pathway categories (Table 4). In the lymph node, spleen, and thymus, 4,848, 4,658, and 3,893 genes respectively were assigned to one or more KEGG pathway categories. For lymph node, spleen, and thymus, the Global overview and maps pathway umbrella category contains the highest number of expressed genes, followed by the Signal transduction category.

The total number of expressed genes enriched in the KEGG pathway Immune system category for lymph node, spleen, and thymus were 266, 221, and 138 genes respectively. The four Immune system pathways that were enriched include Platelet activation pathway, Antigen processing and presentation, B cell receptor signalling pathway, and Intestinal immune network for IgA production. Genes enriched in both Platelet activation and B cell receptor signalling pathways are expressed in lymph node, spleen, and thymus tissues. For the Platelet activation pathway, 87 genes were expressed in all three tissues, including *TLN1*, *TLN2*, *FERMT3*, *ITGB3*, *FGA*, *FGB*, and *FGG* which are involved in the activation and aggregation of platelets. Genes involved in the B cell receptor signalling pathway had 51 genes expressed across all tissues, which include genes such as *SYK*, *PIK3AP1*, *AKT1*,

**Table 3  Categorisation of expressed genes to Gene Ontology terms in lymph node, spleen, and thymus tissues.**

| Gene ontology | Number of genes | | |
|---|---|---|---|
| | Lymph node | Spleen | Thymus |
| **Biological process** | | | |
| Cellular process | 4,429 | 4,239 | 3,551 |
| Metabolic process | 3,793 | 3,634 | 3,155 |
| Localization | 1,152 | 1,101 | 931 |
| Response to stimulus | 1,216 | 1,150 | 856 |
| Developmental process | 1,067 | 983 | 786 |
| Biological regulation | 1,064 | 1,006 | 758 |
| Cellular component organization or biogenesis | 907 | 875 | 772 |
| Multicellular organismal process | 790 | 732 | 526 |
| Immune system process | 611 | 573 | 414 |
| Biological adhesion | 284 | 270 | 209 |
| Reproduction | 186 | 166 | 118 |
| Locomotion | 92 | 92 | 73 |
| Rhythmic process | 10 | 8 | 5 |
| Growth | 4 | 4 | 4 |
| Cell killing | 6 | 5 | 3 |
| **Molecular function** | | | |
| Catalytic activity | 2,972 | 2,856 | 2,411 |
| Binding | 2,957 | 2,806 | 2,369 |
| Transporter activity | 630 | 589 | 447 |
| Receptor activity | 561 | 518 | 338 |
| Structural molecule activity | 502 | 463 | 407 |
| Signal transducer activity | 186 | 170 | 115 |
| Translation regulator activity | 32 | 32 | 31 |
| Antioxidant activity | 19 | 15 | 14 |
| Channel regulator activity | 17 | 16 | 11 |
| **Cellular component** | | | |
| Cell part | 2,339 | 2,247 | 2,002 |
| Organelle | 1,458 | 1,397 | 1,247 |
| Membrane | 957 | 914 | 697 |
| Macromolecular complex | 774 | 761 | 686 |
| Extracellular region | 392 | 365 | 247 |
| Extracellular matrix | 125 | 113 | 79 |
| Cell Junction | 61 | 59 | 43 |
| Synapse | 35 | 36 | 28 |

*AKT2*, *AKT3*, *CHUK*, *IKBKB*, and *IKBKG* that also take part in the PI3K-Akt signalling pathway and NF-kappa B signalling pathway. Genes enriched in the Antigen processing and presentation pathway were expressed in both lymph node and spleen tissues with 61 and 60 genes respectively. All 60 lymph genes expressed in the Antigen processing and presentation pathway were co-expressed in spleen, with genes involved in the MHC I

**Table 4** KEGG pathway term distribution of expressed genes in lymph node, spleen, and thymus tissues.

| | Number of genes | | |
|---|---|---|---|
| **KEGG pathway terms** | Lymph node | Spleen | Thymus |
| **Metabolism** | | | |
| **Global overview and maps** | **947** | **903** | **776** |
| mcf01100:Metabolic pathways | 947 | 903 | 776 |
| **Carbohydrate metabolism** | **0** | **0** | **89** |
| mcf00562:Inositol phosphate metabolism | 0 | 0 | 52 |
| mcf00620:Pyruvate metabolism | 0 | 0 | 37 |
| **Energy metabolism** | **145** | **141** | **136** |
| mcf00190:Oxidative phosphorylation | 145 | 141 | 136 |
| **Glycan biosynthesis and metabolism** | **0** | **0** | **36** |
| mcf00510:N-Glycan biosynthesis | 0 | 0 | 36 |
| **Genetic information processing** | | | |
| **Folding, sorting and degradation** | **0** | **0** | **185** |
| mcf03018:RNA degradation | 0 | 0 | 61 |
| mcf04141:Protein processing in endoplasmic reticulum | 0 | 0 | 124 |
| **Environmental information processing** | | | |
| **Membrane transport** | **39** | **39** | **0** |
| mcf02010:ABC transporters | 39 | 39 | 0 |
| **Signal transduction** | **531** | **439** | **605** |
| mcf04010:MAPK signalling pathway | 194 | 191 | 164 |
| mcf04012:ErbB signalling pathway | 71 | 70 | 67 |
| mcf04014:Ras signalling pathway | 0 | 0 | 142 |
| mcf04015:Rap1 signalling pathway | 0 | 0 | 140 |
| mcf04064:NF-kappa B signalling pathway | 72 | 72 | 0 |
| mcf04310:Wnt signalling pathway | 110 | 106 | 92 |
| mcf04668:TNF signalling pathway | 84 | 0 | 0 |
| **Signalling molecules and interaction** | **77** | **74** | **0** |
| mcf04512:ECM-receptor interaction | 77 | 74 | 0 |
| **Organismal system** | | | |
| **Development** | **102** | **100** | **93** |
| mcf04360:Axon guidance | 102 | 100 | 93 |
| **Immune system** | **266** | **221** | **138** |
| mcf04611:Platelet activation | 107 | 104 | 87 |
| mcf04612:Antigen processing and presentation | 61 | 60 | 0 |
| mcf04662:B cell receptor signalling pathway | 57 | 57 | 51 |
| mcf04672:Intestinal immune network for IgA production | 41 | 0 | 0 |
| **Nervous system** | **86** | **178** | **90** |
| mcf04722:Neurotrophin signalling pathway | 0 | 95 | 90 |
| mcf04725:Cholinergic synapse | 86 | 83 | 0 |
| **Endocrine system** | **208** | **203** | **303** |
| mcf04910:Insulin signalling pathway | 0 | 0 | 99 |

**Table 4** (*continued*)

| KEGG pathway terms | Number of genes | | |
| --- | --- | --- | --- |
| | Lymph node | Spleen | Thymus |
| mcf04915:Estrogen signalling pathway | 83 | 81 | 74 |
| mcf04917:Prolactin signalling pathway | 0 | 0 | 50 |
| mcf04921:Oxytocin signalling pathway | 125 | 122 | 0 |
| mcf04919:Thyroid hormone signalling pathway | 0 | 0 | 80 |
| **Cellular processes** | | | |
| **Transport and catabolism** | **0** | **68** | **325** |
| mcf04142:Lysosome | 0 | 0 | 90 |
| mcf04144:Endocytosis | 0 | 0 | 172 |
| mcf04146:Peroxisome | 0 | 68 | 63 |
| **Cellular community—eukaryotes** | **174** | **172** | **151** |
| mcf04510:Focal adhesion | 174 | 172 | 151 |

(*PSME1*, *PSME2*, *PSME3*, *TAP1*, *TAP2*) and MHC II (*CD74*, *LGMN*, *CTSS*) pathways. The Intestinal immune network for IgA production pathway was specifically enriched in lymph node, with 41 genes enriched in this pathway including genes such as *IL5*, *IL6*, *IL10*, *TGFB1*, *TNFSF13*, and *AICDA* which are involved in the proliferation of B cells into IgA + B cells.

Thymus has the highest number of pathways that are specifically enriched with genes. The enriched pathways include Inositol phosphate metabolism, Pyruvate metabolism, N-Glycan biosynthesis, RNA degradation, Protein processing in endoplasmic reticulum, Ras signalling pathway, Rap1 signalling pathway, Insulin signalling pathway, Prolactin signalling pathway, Thyroid hormone signalling pathway, Lysosome, and Endocytosis. A group of related pathways were observed to be involved in the regulation of the immune system, which include N-Glycan biosynthesis, Protein processing in endoplasmic reticulum, Ras signalling pathway, Rap1 signalling pathway, and RNA degradation pathway). The TNF signalling pathway was the only tissue-specifically enriched pathway in lymph node, with a total of 84 genes enriched in the pathway. Enriched genes such as *TNFRSF1A*, *TNFRSF1B*, *MAP3K7*, *TAB1*, *JUN*, *TRAF1*, *TRAF3*, *BIRC2* and *BIRC3* directly participate in the activation of leukocytes, inflammatory cytokines, and cell survival. All KEGG pathway figures are located in Figs. S7–S15.

## Genes involved in immune-related pathways

The Platelet activation pathway is activated by the adhesion of ADP to purinergic receptor P2Y that begins the complement and coagulation cascade involving genes *TLN2*, *FERMT3*, *ITGB3*, *FGA*, *FGB*, and *FGG* at the end of the cascade. *TLN2* codes for the cytoskeleton protein Talin-2 and is a component that links actin cytoskeleton with integrin and catalyses focal adhesion signalling pathways (*Zhang et al., 2008*). *FERMT3*, together with *TLN1* and *TLN2*, activates integrins beta-3 (*ITGB3*), and when activated, triggers platelet-platelet interactions via binding of fibrinogens alpha, beta, and gamma. These fibrinogens (respectively encoded by *FGA*,*FGB*, and *FGG*) interact to form a polymerisation that form a fibrin matrix that physically plugs ruptured endothelial surface.
In the B cell receptor signalling pathway, a response to foreign antigen binding to the B cell receptor (BCR) is for the gene *SYK* to encode enzyme tyrosine kinase. In effect, tyrosine kinase activates the phosphatidylinositol 3′-kinase (PI3K)-Akt signalling pathway via the BCAP signalling adapter encoded by *PIK3AP1*. The PI3K-Akt signalling pathway then activates the AKT kinase coded by *AKT1*, *AKT2*, and *AKT3* that functions to ensure cell survival by reducing oxidative stress acting on the cell, thus preventing cell apoptosis. AKT kinase also activates the Nuclear factor-kappa B (NF-kappa B) signalling pathway via the IKK enzyme complex (coded by *CHUK*, *IKBKB*, and *IKBKG*) which activate genes that further ensures cell survival (*Faissner et al., 2006*).

In the MHC I pathway, genes *PSME1*, *PSME2*, and *PSME3* encode for proteasome activator PA28, an immunoproteasome involved in the Antigen processing and presentation pathway. Genes *TAP1* and *TAP2* encode for antigen transporters 1 and 2 respectively, and is located on the membrane of the endoplasmic reticulum of a cell. PA28 activator complex play a role in presenting antigens to the antigen transporters, which antigen transport the antigens from cytoplasm to the endoplasmic reticulum to be presented to the MHC I molecules. The *CD74* gene is significant in the MHC II pathway for its antigen processing regulatory function. It codes for HLA class II histocompatibility antigen gamma chain protein that binds to the invariant chain (Ii) polypeptide to form the MHC II protein complex. The complex is then migrated to the endoplasmic reticulum of the cell where the complex is cleaved by legumain (coded by *LGMN*) and cathepsin S (coded by *CTSS*) to a smaller and more stable form of MHC II called CLIP that is ultimately presented on the cell surface.

In the Intestinal immune network for IgA production pathway, as MHC II proteins recognise and present antigens to the A proliferation-inducing ligand (APRIL) receptor (coded by *TNFSF13*), a cascade begins that ultimately differentiate B cells into immunoglobulin A+ B cells. Cytokines interleukin-5, interleukin-6, and interleukin-10 (coded by *IL5*, *IL6*, and *IL10* respectively) together with transforming growth factor beta-1 (coded by *TGFB1*) regulate the proliferation of B cells and eosinophils. The *AICDA* gene that codes for activation induced cytidine deaminase enzyme, also facilitates the class-switch of B cells into IgA+ B cells.

## Lymph node-specific pathway

Two pathways are enriched in the TNF signalling pathway involving genes *TNFRSF1A* and *TNFRSF1B* that code for the TNFR1 and TNFR2 receptors respectively. Each receptor, when bound with their respective ligands activate two separate pathways with different functions. When the TNFR1 receptor is activated, a signalling cascade activates the MAPK signalling pathway cascade which is mediated by the expression of the TAK1-TAB1 kinase complex coded by genes *MAP3K7* and *TAB1*. The activation of the MAPK signalling pathway ultimately regulates the expression of Transcription factor AP-1 (coded by *JUN*), which is known to play a role in apoptosis, cytokine regulation, and leukocyte activation (*Foletta, Segal & Cohen, 1998*; *Kappelmann, Bosserhoff & Kuphal, 2014*). The activation of the TNFR2 receptor by the ligand LTA activates a signalling cascade that is mediated by TNF receptor associated factors 1 and 3 (coded respectively by *TRAF1* and *TRAF3*). The

TNF receptor associated factors then recruit Apoptosis inhibitors 2 and 3 (coded by *BIRC2* and *BIRC3*) to ensure cell survival (*Wang et al., 2012*).

## Thymus-specific immune-related pathways

Out of the three tissues, thymus has the highest number of pathways that are specifically enriched with genes. A total of 984 genes were distributed across 12 pathway categories as analysed with DAVID Bioinformatics Resources using the *Macaca fascicularis* genome reference as the background. Five immune-related pathways were identified, which include Ras signalling pathway, Rap1 signalling pathway, RNA degradation, N-Glycan biosynthesis pathway, and Protein processing in endoplasmic reticulum. Two enriched pathway categories, the Ras signalling pathway and Rap1 signalling pathway, each had 142 and 140 genes expressed in their pathways. The Ras signalling pathway functions as a binary molecular switch to regulate cellular processes such as cell proliferation, migration, differentiation, or growth. Several other pathways are activated by the Ras pathway, and based on the enrichment analysis of the expressed macaque thymus genes, the Mitogen-activated protein kinase (MAPK) cascade and the Rap1 signalling pathway were both activated by the Ras signalling pathway. The Rap1 signalling pathway also acts as a molecular switch to activate cellular processes including the MAPK cascade. These linked pathways suggest a role in the proliferation and differentiation of T cells in the thymus, as the T cell receptor signalling pathway precedes both the Ras and Rap1 signalling pathways. The expressed genes as illustrated in Figs. S12 and S13 suggest that the T cells were undergoing proliferation and differentiation in the individual macaques, however the lack of expressed genes enriched in the T cell receptor signalling pathway and also the lack of the activation of $\alpha\beta T$ cell receptor (TCR) in both Ras and Rap1 signalling pathways further suggests that the proliferation and differentiation of the T cells was not caused by interactions with foreign antigens. The Ras signalling pathway is also known to activate apoptosis functions in a cell, and based on the expression of genes *RASSF1*, *RASSF5*, and *STK4* in the Ras signalling pathway, it is suggested that the apoptosis function is switched on and likely functions in the cell death of thymocytes that fail the positive and negative selections during the developmental process.

The enrichment of RNA degradation pathway suggests the upregulation of eukaryotic exosomes in the macaque transcriptome. Exosomes have multiple roles in the cellular environment, including cell–cell communication, cellular waste removal, quality control of nuclear RNA, and antiviral defence (*Schmid & Jensen, 2008*). Exosomes have been shown to be released by dendritic, B, and T lymphocytes during adaptive and innate immune responses (*Alenquer & Amorim, 2015*). The multiprotein eukaryotic core exosome complex is coded by genes *EXOSC1*, *EXOSC2*, *EXOSC3*, *EXOSC4*, *EXOSC4*, *EXOSC6*, *EXOSC7*, *EXOSC8*, and *EXOSC9*. Co-factors associated with the RNA exosome that are involved in RNA degradation include the Nuclear exosome co-factor, RRRP44 (coded by *DIS3*) and the Cytoplasmic exosome co-factor DIS3 Like Exosome 3′–5′ Exoribonuclease (coded by *DIS3L*). These co-factors play a role in degrading specific target RNAs, for instance, *DIS3L* activates the RNA exosome complex to degrade unstable mRNAs in the cytoplasm (*Staals et al., 2010*), while *DIS3* functions as surveillance for aberrant RNAs (*Molleston et al.,*
*2016*). The TRAMP complex consists of proteins poly(A) polymerase (coded by *PAPD5*, *PAPD7*), zinc-knuckle putative RNA-binding protein (coded by *ZCCHC7*), and RNA helicase (coded by *SKIV2L2*), and is an exosome coactivator complex that interacts with exosomes in the nucleus of eukaryotic cells to initiate 3′ end processing and degradation of ribosomal RNA and small nucleolar RNA (*Jia et al., 2011*).

Other immune-related pathways include the N-Glycan biosynthesis pathway that controls the differentiation of immune-related glycoproteins. Proper folding of glycoproteins such as antigen receptors (T and B cell receptors) and MHC molecules occur in the endoplasmic reticulum and correctly folded proteins are out of the ER via golgi body. The folding and transport of proteins involved in the differentiation of immune cells occur in the Protein processing in the endoplasmic reticulum pathway. Folding of protein products are promoted by the protein Calreticulin, encoded by the gene *CALR* (*Nauseef, McCormick & Clark, 1995*), while the selection and transport of proteins out of the ER is encoded by genes such as *LMAN1*, *PREB*, and *SAR1A* (*Weissman, Plutner & Balch, 2001*; *Nufer et al., 2003*; *Watson et al., 2006*).

## Tissue-specific genes

Venn diagram comparisons for all expressed genes illustrates the overlap of expressed genes among the lymph node, spleen, and thymus tissues (Fig. 1). In the lymph node tissue, a total of 2,630 genes were lymph node-specific genes, with *CLEC4G*, *CCL20*, *SHOX2*, *SHISA3*, and LOC102127387 being the top five most expressed lymph node-specific genes. For the spleen tissue, 253 genes were spleen-specific genes, with *PPBP*, *NKX2-3*, *TCF21*, *CAMK2N1*, and LOC102119108 being the top five most expressed spleen-specific genes. Thymus tissue has 279 thymus-specific genes with *PRB3*, *KLK1*, *ZG16B*, *HTN1*, LOC102142723 as the top five most abundantly expressed thymus-specific genes. A complete list of tissue-specific genes in their respective tissues are listed in Table S4.

## Top highly expressed tissue-specific genes for each tissue

In the lymph node tissue, *CLEC4G*, *CCL20*, *SHOX2*, *SHISA3*, and LOC102127387 are the top five most expressed lymph node-specific genes. Gene *CLEC4G* is expressed in liver and lymph node sinusoidal endothelial cells, and codes for C-type lectin domain family 4 member G protein. Previous studies have identified the protein as a glycan-binding receptor, and is suggested to be involved in cell–cell adhesion processes and acts as a receptor for antigen clearance in lymph nodes (*Liu et al., 2004*). *CCL20* codes for chemokine C-C motif ligand 20 that acts as a ligand for chemokine C-C receptor CCR6. The CCL20-CCR6 ligand–receptor pair is responsible for the chemotaxis of lymphocytes, neutrophils, proinflammatory IL17 producing helper T-cells (Th17), and the regulatory T-cells in response to inflammation (*Frick et al., 2016*). *SHOX2*, a gene which codes for Short stature homeobox protein 2, is a transcriptional factor involved in various embryological development processes (*Branchi et al., 2016*), and is implicated in Turner syndrome, whereby gene knockout of *SHOX2* in mice results in skeletal growth abnormalities and short stature phenotypes (*Gu et al., 2008*). *SHISA3*, which codes for protein shisa-3 homolog, is a member of the Shisa family of proteins that modulate WNT and FGF signalling in developmental processes (*Chen et al., 2014*).

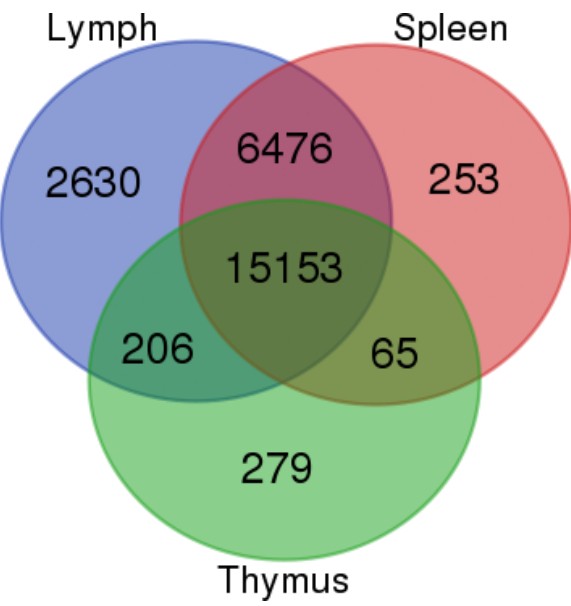

**Figure 1** **Venn diagram of the number of expressed genes (normalised expression values > 1) in their respective tissues.** A total of 15,153 genes were expressed in all three lymph node, spleen, and thymus tissues. The number of tissue-specific genes for lymph node, spleen, and thymus are 2,630, 253, and 279 respectively.

For the spleen tissue, the top five most expressed spleen-specific genes are *PPBP*, *NKX2-3*, *TCF21*, *CAMK2N1*, and LOC102119108. *PPBP* (also known as *CXCL7*) codes for Pro-platelet basic protein, a platelet-derived growth-factor that acts as a chemoattractant and activator of neutrophils, and is involved in blood clotting and inflammation (*Herring et al., 2015*). Homeobox protein Nkx-2.3 is coded by *NKX2-3*, while Transcription factor 21 is coded by TCF21. Both NKX2-3 and TCCF21 are transcription factors that are suggested to be significant in the organogenesis of the spleen (*Brendolan et al., 2007*; *Czömpöly et al., 2011*). *CAMK2N1* codes for Calcium/calmodulin-dependent protein kinase II inhibitor 1 which inhibits Calcium/calmodulin-dependent protein kinase II (CaMKII) activity pertinent to cell growth processes (*Wang et al., 2008*).

The top five most expressed thymus-specific genes include *PRB3*, *KLK1*, *ZG16B*, *HTN1*, and LOC102142723. *PRB3* gene codes for Proline Rich Protein BstNI Subfamily 3, which acts as a receptor for Gram-negative bacterium *Fusobacterium nucleatum* (*Gillece-Castro et al., 1991*). KLK1 codes for Kallikrein Serine Protease 1, and is suggested to have an anti-inflammatory role in the adaptive immune system by promoting lymphocyte proliferation (*Panos et al., 2014*). The gene *ZG16B* codes for Zymogen Granule Protein 16B, and functions to regulate chemokine CXCR4 and acts as an endogenous ligand of Toll-like receptor 2 (TLR2) and TLR4 for the expression of chemokines CCL5 and Macrophage migration inhibitory factor (MIF) (*Park et al., 2011*). HTN1 codes for protein Histatin-1, and is involved in antifungal and antibacterial activities (*Oppenheim et al., 2007*).

## Validation of RNA-seq differential gene expression

Twenty four genes were selected to validate the differential gene expression data obtained from the RNA-Seq platform. Fold change values from RNA-Seq platform were compared with fold change values obtained from the NanoString nCounter XT platform. Figure 2 provides a graphical comparison of $\log_2$ transformed fold change values between the two platforms. The concordant fold change directions of all genes confirms the differential gene expression results from the RNA-Seq platform. The NanoString gene expression assay utilises barcoded probes that hybridise to their specific RNA target and are directly counted via the colour-coded probes. This eliminates the need to enzymatically amplify the template and also minimises pipetting procedures to reduce errors and contaminations. The platform reliably captures gene expression profiles effectively with relatively small amounts of template RNA (100 ng) and also serve as an alternative to RT-qPCR gene expression assays (*Radke et al., 2014*; *Hu et al., 2016*). Previous cynomolgus macaque studies utilised RT-PCR and RT-qPCR to validate their high-throughput sequencing results, and as such, the present study is the first to validate cynomolgus macaque RNA-Seq data using the NanoString nCounter XT gene expression assay. The platform's relative ease of use, quick turnaround, reproducibility, and sensitivity make it suitable for medium to high-throughput screening of panels of genes for future biomedical science research utilising the cynomolgus macaque as NHP models.

## DISCUSSION

This manuscript presents the lymph node, spleen, and thymus transcriptome of wild Malaysian cynomolgus macaque sequenced with the Illumina HiSeq 2500 platform. A combined total of 174,208,078 reads were obtained from the HiSeq 2500 sequencing run of the three tissues with three replicates each. Reference-guided mapping of the sequencing reads to the *M. fascicularis* reference genome revealed mapping percentages ranging from 53.00% to 63.00%. To determine whether the low mapping percentage was caused by contaminants, the unmapped reads from M1, M2, and M5 Lymph were BLAST against NCBI's nucleotide (nt) database and classified into their specific taxa. The majority of the reads were found to map to Cercopithecidae and other primate families, while the remaining reads either did not have any BLASTN hits or were unassigned (Fig. S16). This suggests the presence of novel transcripts within the lymph node, spleen, and thymus transcriptome that are yet to be described or annotated in the *M. fascicularis* reference genome, indirectly reflecting the incompleteness of the genome's annotations (*Zhao & Zhang, 2015*).

Previous transcriptome sequencing endeavours via high-throughput sequencing methods were conducted on laboratory-bred cynomolgus macaques that originated from Vietnam, China, and Mauritius (*Huh et al., 2012*; *Pipes et al., 2013*; *Peng et al., 2014*). Our sequencing results show the expression of 253 spleen-specific genes, contrasting with *Huh et al. (2012)* whereby no spleen-specific genes were called in their sequencing endeavour. Such an occurrence is likely due to the lower total sequencing reads obtained in *Huh et al.*'s (*2012*) sequencing effort, amounting to 4 million reads across 16 different tissue libraries

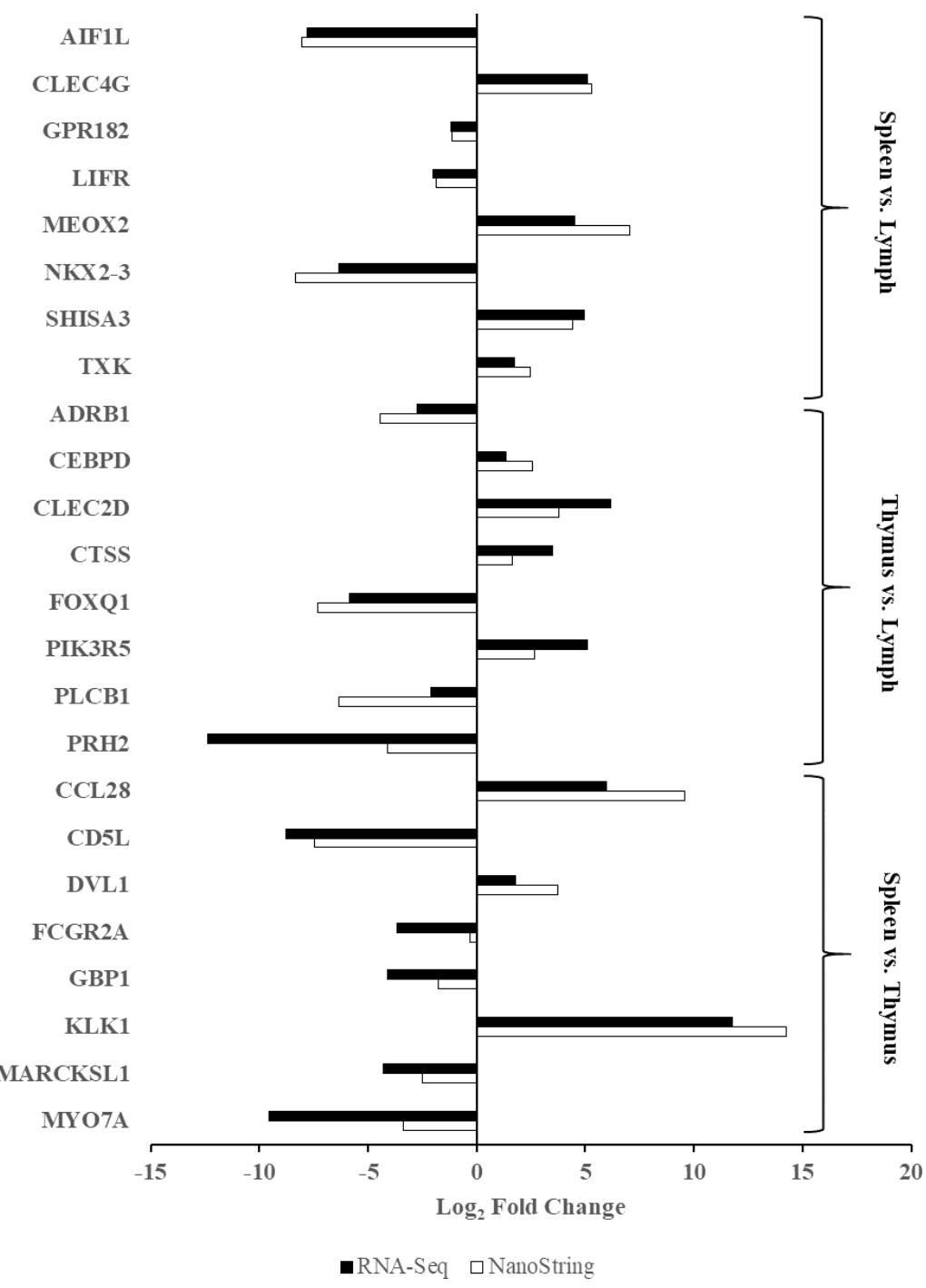

**Figure 2  Validation of RNA-Seq differential gene expression results.** Log$_2$ transformed fold change values from three sets of tissue comparisons (Lymph vs. Spleen; Lymph vs. Thymus; Spleen vs. Thymus) were obtained from RNA-Seq and NanoString nCounter XT platforms. Solid bar represents fold change values obtained from RNA-Seq platform, while striped bar represents fold change value obtained from NanoString nCounter XT platform. Concordance of directionality of fold change values between the two platforms confirms the RNA-Seq differential gene expression results.

from two individuals compared with this study's 174.2 million reads across a total of nine tissue libraries from three individuals. An alternative explanation is that the study by *Huh et al. (2012)* contains 16 tissue samples, whereas the present study only contains 3 tissues, and is therefore easier to be tissue specific in the latter case. *Peng et al. (2014)* sequenced a total of 189.4 million reads from the lymph node, spleen, and thymus transcriptomes of Chinese cynomolgus macaque using the Illumina HiSeq 2000 technology. A preliminary comparison of gene expression values (normalised expression values) of 15 tissue-specific genes generated from this study with the gene expression values (normalised sFPKM) of Chinese cynomolgus macaques described in *Peng et al. (2014)* show nine out of 15 gene expression patterns in concordance (Table S5), however the gene expression magnitude of certain genes such as *PRB3* and *ZG16B* vary due possibly to variations in individuals and/or populations, which is evident in primate species (*Enard et al., 2002*; *Hsieh et al., 2003*). In the future, more robust transcriptomic comparisons between the Malaysian cynomolgus macaques and the Chinese and Mauritian cynomolgus macaques sequenced by *Pipes et al. (2013)* and *Peng et al. (2014)* may potentially reveal more insights in the transcriptomic and genetic differences between the three populations.

## Concluding remarks

In an immunological aspect, lymph node, spleen, and thymus transcriptomic data of wild Malaysian cynomolgus macaques generated from this study are potentially valuable for further studies. For AIDS-related immunology studies, the panel of genes expressed in the MHC I and MHC II pathways are advantageous for MHC genotyping studies of the Malaysian cynomolgus macaque population, which was suggested to have relatively higher nucleotide diversity compared with other populations of cynomolgus macaques in Southeast Asia (*Smith, McDonough & George, 2007*). Additional screenings of MHC genotypes in Malaysian cynomolgus macaques may aid in further understanding and discovering novel immune regulation mechanisms and responses. Previous studies have characterised MHC genotypes as well as their effects on SIV progression in cynomolgus macaques (*Burwitz et al., 2009*; *Campbell et al., 2009*; *Aarnink et al., 2011*; *Borsetti et al., 2012*). The other benefit of transcriptomic data obtained from wild populations is how they more closely reflect the 'wild' and variable environment humans undergo on a daily basis. It is notable that in controlled laboratory environments, the absence of biotic and abiotic pressures associated with natural environments is unrepresentative of how immune responses function in the wild in response to pathogenic infections and injury. In assessing natural immune responses, both genetic and environmental factors play concerted roles. The transcriptomic information obtained from the three immune organs provides a fundamental outlook to the immune responses enriched in the wild cynomolgus macaque. Immunology data gleaned from wild cynomolgus macaque populations will benefit researchers in identifying significant immune responses and phenotypes that are translatable to natural human settings (*Pedersen & Babayan, 2011*).

## ACKNOWLEDGEMENTS

We would like to thank the Department of Wildlife and National Parks (DWNP), Malaysia, for the permission to conduct this study. We would also like to thank Dr. Ng Wei Lun (School of Life Sciences, Sun Yat-sen University), Nancy Liew Woan Charn (Laboratory of Vaccines and Immunotherapeutics, Institute of Bioscience, Universiti Putra Malaysia), and Mok Shao Feng for their invaluable guidance and assistance throughout the preparation of this manuscript.

### Funding

This research was supported by Research University Grant Scheme (Project No: GP-IPB/2013/9413602) provided by Universiti Putra Malaysia, Malaysia. The funders had no role in study design, data collection and analysis, decision to publish, or preparation of the manuscript.

### Grant Disclosures

The following grant information was disclosed by the authors:
Research University Grant Scheme: GP-IPB/2013/9413602.
Universiti Putra Malaysia, Malaysia.

### Competing Interests

The authors declare there are no competing interests.

### Author Contributions

- Joey Ee Uli conceived and designed the experiments, performed the experiments, wrote the paper, prepared figures and/or tables.
- Christina Seok Yien Yong conceived and designed the experiments, analyzed the data, contributed reagents/materials/analysis tools, reviewed drafts of the paper.
- Swee Keong Yeap conceived and designed the experiments, analyzed the data, reviewed drafts of the paper.
- Jeffrine J. Rovie-Ryan contributed reagents/materials/analysis tools.
- Nurulfiza Mat Isa and Noorjahan Banu Alitheen analyzed the data, reviewed drafts of the paper.
- Soon Guan Tan analyzed the data, contributed reagents/materials/analysis tools, reviewed drafts of the paper.

### Animal Ethics

The following information was supplied relating to ethical approvals (i.e., approving body and any reference numbers):

The sampling and culling of *M. fascicularis* macaques were carried out by the Department of Wildlife and National Parks (DWNP), and were performed according to the guidelines set by the Institutional Animal Care and Use Committee (IACUC), University of California,

Davis, United States of America. Permission to obtain macaque lymph, spleen, and thymus tissue samples for research purposes by the Genetics Lab at the Department of Biology was granted by DWNP. Approval number JPHL & TN (IP): 80-4/2.

## Data Availability

Ee Uli, Joey; Yong, Christina Seok Yien; Keong Yeap, Swee; Banu Alitheen, Noorjahan; Rovie-Ryan, Jeffrine J; Isa, Nurulfiza Mat; Guan Tan, Soon (2017): RNA-sequencing of Lymph, Spleen, and Thymus Transcriptome of Peninsular Malaysia *M. fascicularis*. figshare. https://doi.org/10.6084/m9.figshare.4697539.v2.

## Supplemental Information

Supplemental information for this article can be found online at http://dx.doi.org/10.7717/peerj.3566#supplemental-information.

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
