# Peer review of "RNA sequencing (RNA-Seq) of lymph node, spleen, and thymus transcriptome from wild Peninsular Malaysian cynomolgus macaque (Macaca fascicularis)"

_PeerJ, doi:10.7717/peerj.3566_

## Round 0.1 · original submission · Major Revisions

The reviewers have provided extensive and detailed comments regarding robustness and the validity of some aspects of the methodology. All these concerns need to be addressed before further consideration of the manuscript.

Reviewer 1 ·

Basic reporting

* Supplementary tables(Table S1-S3) are missing for review.
* The search of SRP096937 on NCBI SRA doesn't return any result(https://www.ncbi.nlm.nih.gov/sra/?term=SRP096937).
* Reporting of "Total of 174,208,078 raw sequencing reads" in line 44 or "the number of reads for lymph, spleen, and thymus tissues... 47,559,293, 60,285,505, and 56,583,352" in line 278 is useful for conveying the general sense of the size of the data, but it would also be informative if Uli et al. report the number of reads of each of individual libraries as well as a supplementary table.
* line 109: Uli et al. indicate that previous studies were performed using healthy individuals raised in laboratory conditions. Does this mean the animals used in this study are not? The authors mention in the following sentences the limitation of laboratory animals but do not clearly make a statement about the animals in the study.
* line 264: What does FOV mean?
* In Table 3, what does "mcf" stand for?
* In all the figures in Supplementary materials, labels in the figures are difficult to see.
* line 505: what does the word "sFPKM" "mean? is this a typo of "FPKM"?
* Pie charts are heavily used for figure 1, 2, 3, and they contain both technical and aesthetical problems: 1) It is very difficult to distinguish pie charts across the tissues(they look almost identical 2) in some pie charts, there are overlaps of labels(developmental process and response to stimulus in line 345, for example).

Experimental design

* line 488: Can you provide the data for the part "The "majority" of the reads were found to map to Cercopithecidae"?
* line 517: In the section "Top Highly Expressed Tissue-Specific Genes for Each Tissue", authors describe five most expressed genes for each tissue. Where can I see the mean FPKM and standard deviation for these genes?
* line 217 : Indicate the scatter plot that is being referred to.
* S5-S6 : Why do thymus samples have wiskers that are drastically different from those of lymph nodes and spleen?
* S1-S6 : In all figures, samples were compared in a pairwise manner only. Why not including all samples at the same time?

Validity of the findings

* In the manual page of Empirical analysis of DGE(http://resources.qiagenbioinformatics.com/manuals/clcgenomicsworkbench/803/index.php?manual=Empirical_analysis_DGE.html), it states "the test is applicable to count data only". However, Uli et al. computed expression levels as FPKM, which is not appropriate for EDGE test. Please explain why you used this method.
* Based on S1-S6, it seems that normalization was performed in a pairwise manner, and as a result, normalized FPKM values for a particular sample will be different depending on what tissues they are being compared to. In line 237, how did the authors impose FPKM > 1 as a threshold for expression? The second question is why FPKM > 1 is used as opposed to FPKM > 0.5, FPKM > 0.8, FPKM > 2, or FPKM > 5, etc?
* line 496: Here the authors indicate their results regarding spleen-specific genes are different from those of Huh et al. and attribute the reason to the differences of depths of the libraries. An alternative explanation is that the study by Huh et al. contains 16 samples, where the present study only contains 3 tissues, therefore it is easier to be tissue specific in the latter case.

Additional comments

* line 92: Insert a space before the word "High".
* In line 36, 262, 263, and 683, the word "utilised" is used but in line 79,202, and 242, the word "utilized" is used.
.* line 139: Remove a space between the word "tissues" and an em-dash.
* line 135: Insert a space between existing and cynomolgus.
* line 151: Insert a space between "of" and "conflict".
* line 151 and 16: the usage of the word "conflict" is not clear. It seems that the authors used this word to indicate a "group" of macaques, but the list of collection nouns for monkeys in Wikipedia(https://www.wikiwand.com/en/List_of_English_terms_of_venery,_by_animal) does include the word. The ones listed in Wikipedia are barrel, cartload, tribe, troop, or wilderness.
* line 162: Insert ", respectively" after 2.11kg.
* line 178: Insert a space between the word "Partition" and &.
* line 311: Insert a comma between "genes" and "respectively"
* line 323: Change "CLCGW" to "CLC Genome workbench".
* line 389: Change IL-5 to IL5 and IL-10 to IL10
* line 151, ofconflict --> of conflict
* line 182, rRNa --> rRNA
* line 230, The "p-value < 0.05" seems redundant since authors state the p-value is also corrected with FDR.

Reviewer 2 ·

Basic reporting

The text presented in Uli et al requires serious improvement to the writing prior to publishing. There are numerous grammatical, punctuation, spacing, and word choice usage issues that must be addressed. The entire text should be reviewed, particularly where emphasis through sematic usage was attempted. The edits were too many to list here but issues that effect the understanding of the text are as follows:
L50 "and cell part." - Clarify
L51-52 "and thymus fall under Global overview and maps" - Clarify Global overview
L174 "is the only and most widely distributed macaque" - If it is the only the assumption is 'most' widely distributed.
L122 "best of interest" - awkward usage, try "therefore necessary"
L151,160 "ofconflict" - review, meaningingless.
L161 "sub-adults" - awkward usage try "adolescents"

The citations I reviewed were well founded.

The structure of the article needs improvement. The abstract is particularly verbose and cumbersome to read. Consider redacting significantly the methods and validation results as the process isn't the purpose of the study, the data is. Consider moving figures 1-3 to a supplemental section or convert to a table representation. The labeling is of low quality due to placement issues and the value of their visual representation is minimized by the sheer size of and number of panels. Consider moving figures 4-15 to a supplemental section. In their current form they are difficult to read and do not add to the narrative as data. L517 - 688 this section of text belongs in the results section. In general all of the figures were below acceptable publishing standards due to labeling, highlighting color or resolution issues except Fig 13.

This body of work is self-contained and relevant in regards to the stated goals of the paper.

Experimental design

The primary research is within the aims and scope of the journal.

The research question is well defined, relevant and meaningful and the knowledge gap is meticulously described.

The investigation was performed to technical and ethical standards. You need to clarify in the ethical statement the country to which the DWNP reports to in L151-156. It is clarified later in the text and so it is just a text edit at this point.

The methods are described in sufficient detail.

Validity of the findings

Impact and novelty were not assessed, no negative results were reported to be reviewed.

I have concerns about the robustness of the data. First 3 individuals are being summarized in the data as single data points by tissue type. The contribution of each individual should be delineated in a table or the text. The total read count should be sufficient to cover all three samples, however, a weak sample could influence downstream gene reporting and analysis and the data presented are not granular enough to assess. Second, again concerning the individuals, a great deal of the text is spent defending the need for wild caught individuals to understand the genetic diversity available. However, wild caught animals could have a host of health issues that we have no way to assess without them. I feel that clinical findings are necessary to support this data. Were their temperatures recorded? Were they free of obvious signs of disease? Were all 3 individuals localized to a large or small geographic region during sampling? Why were only males selected? Was a similar battery of screenings for common pathogens completed as would be expected of a colony bred animal prior to being used in a study? Lastly a clarification: ~60% mapping. L147. Is that 63% of the known transcriptome or 63% of the reads?

The conclusions are sufficient

I did not identify any unsupported speculation.

Additional comments

I believe this body of work will be useful provided the comments of the previous sections are addressed sufficiently.

---

## Round 0.2 · Minor Revisions

The comments from reviewers are comprehensive and in detail address a few issues with consistency in the manuscript.

Reviewer 1 ·

Basic reporting

- S4-6, the units for the y-axis should be provided. Counts or FPKM? Are they transformed with log2 or log10?
- The brief description should be provided for the table S16. For example, In "Figure_S16_Unmapped-reads_assignment_charts", what are the boxes and what do different colors represent?(add legend in the figure as to what the degree of green color represent) Why are there multiple boxes? Also, relationships between labels and boxes are not so clear(for example, which boxes are associated with Eukaryota and with Bilateria)? In short, tell readers how to parse this diagram.
- L228, L237: Clearly indicate whether normalised fold change is log2 or log10
- Table_S3_Differentially_expressed_genes:
According to the authors, PRB3 is thymus-specific, but in the fold change is reported to be 2201.25 in "Spleen vs. Thymus" sheet whereas -2114.04 in "Lymph vs. Thymus". It seems that the third sheet should have been labeled as "Thymus vs. Spleen" instead of "Spleen vs. Thymus". Furthermore, I suggest that "Normalised fold change" column be clearly labeled so as to indicate the direction of the fold change.
- RNA-Seq Normalised Means (as in Table_S5_Comparison_with_Peng_et_al) should be reported for all genes as an additional supplementary table.
- CLEC4G, SHISA3, and CCL20 are reported to be lymph node-specific. Question is, why aren't CLEC4G and SHISA3 reported in "Lymph vs.Thymus" sheet in Table S3? and CCL20 in "Lymph vs. Spleen" sheet? Similarly, NKX2-3 is reported to be spleen-specific but isn't reported in "Spleen vs. Thymus" sheet in Table S3. Indeed, those genes have a mean expression level equal to 0 in thymus tissues(as reported in Table_S5) and I speculate that those genes were filtered out because their log fold change values are infinity. However, authors should process their data so that those genes are included in their differential gene expression analysis list so that it is consistent with results of the tissue-specific analysis.

Experimental design

Authors addressed the concerns raised in the previous review.

Validity of the findings

Authors addressed the concerns raised in the previous review.

Additional comments

- L613: completeness -> incompleteness
- L622: ofnine -> of nine
- Table S3: what is the unit of Normalised fold change? log2 or log10? Indicate it on the column label.
- L190: normalized -> normalised (for consistency)
- L242, 245: ver. -> version (to be consistent with the usage of version in L238)
- L88: Insert a space before "High throughput..."
- L189: Insert a space before "The final ..."
- L244: Insert a space before "KEGG pathway ..."
- L318: Insert a space before "In Lymph vs. Spleen ..."
- L322: Insert a space before "Table S3 ..."
- L389: Insert a space before "The TNF ..."
- L623: Insert a space before "An alternative ..."
- L625: Insert a space before "Peng et al ..."

Reviewer 2 ·

Basic reporting

No comment.

Experimental design

L160-165 - As indicated in the rebuttal add a comment relating to the animals being free of disease upon visual examination and members appeared to be of the same family group.

Validity of the findings

Thymus M2 and M5 are particularly weak with 1/3 to 1/6 of the reads mapped as compared to other samples. While I do not consider this a risk to the observations made during the aggregate analysis it does lower the resolution as observed in both the MCA and various gene expression data. Adding more sequence data would significantly raise the value of the data set to the research community. While I strongly recommend adding sequence data to these samples I do not consider it a bar to publishing this paper.

---

## Round 0.3 · accepted · Accept

I am sure this paper contributes to better understanding of cynomolgus macaque and other primate animal models.